# Location of Tensile Damage Source of Carbon Fiber Braided Composites Based on Two-Step Method

**DOI:** 10.3390/molecules24193524

**Published:** 2019-09-28

**Authors:** Gang Ding, Chunbo Xiu, Zhenkai Wan, Jialu Li, Xiaoyuan Pei, Zhenrong Zheng

**Affiliations:** 1School of Textiles Science and Engineering, Tianjin Polytechnic University, Tianjin 300387, China; dinggang@tjpu.edu.cn (G.D.); wanzhenkai@tjpu.edu.cn (Z.W.); lijialu@tjpu.edu.cn (J.L.);; 2Supervisory and Evaluation Office, Tianjin Radio & TV University, Tianjin 300191, China; 3School of Electrical Engineering and Automation, Tianjin Polytechnic University, Tianjin 300387, China; xiuchunbo@tjpu.edu.cn; 4Composites Research Institute, Tianjin Polytechnic University, Tianjin 300387, China

**Keywords:** carbon fiber braided composites, location of damage source, two step method, four-point arc method, probabilistic neural network

## Abstract

Acoustic emission (AE) source localization is one of the important purposes of nondestructive testing. The localization accuracy reflects the degree of coincidence between the identified location and the actual damage location. However, the anisotropy of carbon fiber three-dimensional braided composites will have a great impact on the accuracy of AE source location. In order to solve this problem, the time-frequency domain characteristics of AE signals in a carbon fiber braided composite tensile test were analyzed by Hilbert–Huang transform (HHT), and the corresponding relationship between damage modes and AE signals was established. Then, according to the time-frequency characteristics of HHT of tensile acoustic emission signals, the two-step method was used to locate the damage source. In the first step, the sound velocity was compensated by combining the time-frequency analysis results with the anisotropy of the experimental specimens, and the four-point circular arc method was used to locate the initial position. In the second step, there is an improvement of the Drosophila optimization algorithm, using the ergodicity of the chaotic algorithm and congestion adjustment mechanism in the fish swarm algorithm. The smoothing parameters and function construction in the probabilistic neural network were optimized, the number of iterations was reduced, the location accuracy was improved, and the damage mode of composite materials was obtained. Then, the damage location was obtained to achieve the purpose of locating the damage source.

## 1. Introduction

Carbon fiber composites have the characteristics of high strength and modulus, small thermal expansion coefficient and fatigue resistance [1,2,3]. Such composites have excellent performance in crack resistance, impact resistance, and damage tolerance [4]. In the long-term use process, due to the excessive tension operating load, the composite may have debonding, edge damage (including fracture), and other defects [5,6]. Common nondestructive testing methods for composite materials include infrared thermography [7], ultrasonic C-scan technique [8], eddy current testing [9] and penetrant testing [10], etc. These technologies are widely used in the experimental research, device monitoring and failure analysis of composite materials. Although these technologies are relatively mature, they also have the following shortcomings. Firstly, the total volume of the detection device needs to change with the volume of the detected object. For example, the working principle of ultrasonic testing and radiographic testing is to generate ultrasonic and radiographic excitation through an external energy supply device. The existence of defects can be found according to the discontinuity of the corresponding physical properties of the defect of the detected body. When testing composite materials, in order to avoid the occurrence of ultrasonic, radiation cannot penetrate, or signal deviation leading to the receiver cannot receive, the general method is to increase the excitation power and receiver receiving area. This leads to the detection device increases with the increase of the volume of the object being examined, and even requires a specific detection device, which cannot achieve versatility. Secondly, the selection of testing methods has limitations on the material types and defect types. For example, penetration testing can only detect defects on the surface and near the surface of the material. Ultrasound scanning requires a smooth surface of the object to be inspected for placement of an ultrasound probe, etc. Conventional nondestructive testing techniques may not be able to detect specific materials or defects. Thirdly, conventional nondestructive testing techniques need to recognize defect signals only when the defect reaches a certain level. The breakage of fibers in the bundles and the debonding of fibers in a small area cannot be found in time. Compared with damage, detection has hysteresis. Fourth, the detection signal has no characteristic. For fiber reinforced composites, conventional nondestructive testing cannot distinguish whether the damage is fiber breakage or matrix cracking by testing signal. Moreover, it is not possible to quantitatively analyze the percentage of each injury to the total damage, and the detection signal can only reflect the presence of the injury and the extent of the damage.

Acoustic emission (AE) testing is a non-destructive testing technology, which is different from the conventional non-destructive testing technology. It belongs to the AE method without external excitation. Compared with the traditional nondestructive testing technology, AE technology has the following two characteristics. First, the dynamic information of material and structure damage can be detected in real time. It can monitor the generation and propagation of cracks. Secondly, the AE signal detected is emitted by the material itself, rather than the excitation signal introduced by the outside world. Therefore, AE technology has the following advantages. First, it has initiative. For large components, active signal output can reduce the number of detection equipment and save detection time. Through a large number of wiring and a single loading in the early stage, AE signals will be generated at the damaged parts of the components. From this, the location and types of damage can be determined, and the detection efficiency can be improved. Secondly, the AE signal is used to identify the type of damage and the location of damage. The location of the AE signal is based on the location of the signal receiver, but the geometric shape of the workpiece itself is not required. Thirdly, the AE signal is received by the receiver, which can be regarded as the real-time production of the workpiece damage. It can be applied to real-time and rapid monitoring of damage and its development. Fourthly, the signals of different failure modes in the AE signal source can be separated by spectral analysis technology. According to the dominant AE signal, the type and degree of internal damage and failure of the specimens are judged. AE nondestructive testing technology has become a research hotspot for defect and damage detection in composite materials [11,12].

Three-dimensional braided composite materials, first of all, it uses three-dimensional braiding technology to braid reinforced fibers into three-dimensional monolithic fabric (preform). Then, the preform is compounded with the matrix to make composite. The reinforcing fibers in three-dimensional braided composites have a spatial interweaving structure. Therefore, in addition to the advantages of high specific strength and high specific modulus, it also has better impact resistance, higher damage tolerance and energy absorption [13]. During the tensile process of three-dimensional braided composites, part of the strain energy release forms acoustic signals in the form of stress waves [14]. Acoustic emission (AE) technology can acquire the acoustic emission signals of these internal damages [15]. AE technology can be used to establish a corresponding relationship between the collected signals and the development of internal damage of materials [16]. Through a series of signal analysis methods, AE signal denoising, and acoustic source waveform recognition, the damage source location is finally obtained [17].

In the research of AE detection technology, Barile proposed a method to detect and identify different types of damage in carbon fiber composites (CFRP) using AE technology [18]. Nair A et al. used advanced pattern recognition technology to identify failure mechanisms in CFRP reinforced beams by collecting AE data [19]. The correlation between the main failure modes of glass fiber reinforced composites (GFRP) and their AE in tensile test and double cantilever beam (DCB) test was evaluated by Bohmann [20]. Rescalvo et al. proposed a method to detect wood beams modified with CFRP by AE technology [21]. Matvienko et al. developed a program for monitoring degradation processes and predicting the residual strength of polymer composite packaging multilayers [22]. Saeedifar et al. found that, under quasi-static and dynamic transverse loads, AE is a powerful tool for studying the barely visible impact damage (BVID) of laminated composites [23]. Bashkov et al. introduced the method of estimating damage accumulation of GFRP prepared by vacuum and vacuum-high pressure moulding in AE parameter processing technology [24]. Qingwen You et al. carried out drop hammer impact test and small mass impact test on composite laminates, and found that AE signal combined with empirical mode decomposition (EMD) can be used for real-time health monitoring [25]. Shahkhosravi et al. proposed a very promising method based on AE and finite element (FE) techniques to evaluate the initiation and propagation of damage in high-speed drilled composite laminates [26]. Barile et al. tested the ability of AE technology to detect unidirectional delamination of carbon fibre composites under double cantilever loads and found out the possible correlation between the frequency content of acoustic signals and damage evolution [27]. Based on acoustic emission (AE) detection technology in industrial environments, Esola Shane et al. introduced a multi-dimensional assessment method for wing beam components identification of composite fixed-wing aircraft using composite structure and non-destructive assessment method, which can analyze and identify key damage areas [28].

Since there are dozens of characteristic parameters of AE events, there is a certain correlation between them in clustering the analysis of signals, so it is necessary to optimize the characteristics of data. In recent years, with the rapid development of artificial intelligence, neural network has been used to model data. The trained model is used to classify and judge unknown data, which improves the modeling speed and recognition efficiency. Kalafat et al. present an alternative localization method based on the use of neural networks, using experimental training data as a modeling basis. It was shown that the neural-network-based method is not only superior by a factor of six in accuracy, but also results in a lower scattering of the localized source positions by a factor of 11 [29]. Han et al. proposed an extraction method of damage modes in composite laminates from (AE) signal based on ensemble empirical mode decomposition (EEMD) and decorrelation algorithm. The results showed that DEEMD is the more effective solution for extracting all damage modes existing in a single AE signal than EMD, can eliminate mode mixing [30]. Tabrizi et al. used the experimental and numerical investigation on fracture behavior of glass/carbon fiber hybrid composites. Damage occurrence is recorded using AE method and then damage types are classified by means of K-means algorithm. Results showed four clusters of acoustic data corresponding to four failure types, i.e., matrix cracking, interface failure, fiber pullout, and fiber breakage [31]. Fatih et al. used the acoustic emission (AE) registration technique and its location detection capability to identify and locate the damage modes during the tension tests. The k-means ++ algorithm is applied to cluster similar AE events and obtain reliable correlations between the damage modes and AE characteristics. The correlations between the AE clusters and damage modes are validated with the finite element model [32].

The premise of traditional AE source location is to determine the propagation speed of acoustic wave in solid medium, and to determine the location of acoustic emission source by the time difference between the acoustic wave arriving at the two sensors. However, carbon fiber braided composites have anisotropic properties [33]. Sound wave velocity is not only related to material characteristics, thickness and propagation angle, but also to the change of material characteristics caused by environmental factors such as temperature and humidity, thus changing the propagation speed of stress wave and affecting the location results of damage sources [34]. In order to reduce the location error of damage sources, the time-frequency relationship of AE signals is compensated when tensile damage occurs in carbon fiber braided composites. The damage of carbon fiber braided composites is identified and the initial location of damage sources is formed. The chaos algorithm and fish swarm algorithm are used to improve the Drosophila algorithm and improve its recognition accuracy. The key parameters of probabilistic neural network are further optimized, and the improved Drosophila optimization algorithm is used to locate the damage source of braided composites accurately. Through the numerical analysis of the tensile damage signal of carbon fiber braided composites, the results show that the improved Drosophila algorithm optimizes the smoothing parameters and function construction of probabilistic neural network, reduces the number of iterations, and improves the identification and location accuracy of different damage of carbon fiber braided composites.

## 2. Tension Damage Detection Test of Carbon Fiber Braided Composites

### 2.1. Materials and Equipment

In this paper, carbon fibre braided composites by four-step 1 × 1 three-dimensional four-directional braiding process are taken as the research object for tensile test. The braided yarn of the test sample is T700−12K carbon fiber with a density of 1.76 g/cm^3^ and a linear density of 0.8 g/m. Its technological parameters are as follows: surface braiding angle is 22.60°; internal braiding angle is 32.14°; pitch length is 6.0 mm; fiber volume content is 45.00%; size is 250 mm × 25 mm × 4 mm. The internal structure sketch and physical diagram of three-dimensional four-directional carbon fiber braided composites are shown in Figure 1.

Before the tensile test, the sound velocity was determined by the standard lead-breaking test at the position near the sensor. When the measured sound wave amplitude is greater than 85 db, it is considered to meet the requirements. In the tensile test, the AE signals during the tensile process are obtained by AE testing technology (SAEU2S Acoustic Emission Signal Acquisition System), and the corresponding mechanical properties parameters are obtained by Shimadzu Universal Material Testing Machine for analysis. To avoid testing error, specimens with each parameter were tested five times. The tensile process of a 3D braided composite is shown in Figure 2.

### 2.2. Analysis of the Relationship between Mechanical Properties and Acoustic Signals

In the process of tensile test, mechanical properties such as tensile stress-strain curve and acoustic signal parameters such as energy, ringing count, and RMS voltage can be collected. The load-displacement curves of the specimens correspond to the energy parameters of acoustic emission signals as shown in Figure 3.

From the load/KN-displacement/mm (red) curve in Figure 3, it can be seen that at the beginning of the tensile process, the tensile stress along the tensile direction of the sample increased rapidly, and then gradually presented a linear trend until the final fracture. The non-linear change in the initial stage may be caused by the micro-defects in the experimental specimen, such as impurities and voids in the material. With the process of tension, the effect of internal defects on the specimen becomes smaller and smaller.

It can be seen from the energy/eu-displacement/mm (blue) curve in Figure 3, as the tensile load increases, the failure of the samples were also accumulating. Energy release occurs in all time periods, the energy parameters of AE signals can clearly distinguish the tensile process of the sample. Different source signals existed in sequence of time. There were several important mutation points, which indicated that energy was released centrally after a long period of accumulation. When these energies were released centrally, it was also the time when the crack of the sample changed strongly. That was to say, at the critical loading point where the crack of the sample increased to the next stage, the release of energy in the AE signal increased rapidly.

In Figure 3, according to the energy and amplitude changed of AE signal peak parameters, load-displacement curves, and crack growth patterns, the tensile process of samples can be basically divided into three stages: Section 1, micro-crack generation (initial damage); Section 2, micro-crack propagation (damage evolution); and Section 3, instability failure (damage and failure) stages. In the micro-crack generation stage, with the increased of tension load and displacement, the energy parameters of AE signal continued to accumulate, and the load curve gradually increased. This indicated that a small amount of damage occured with the increase of load in the initial stage of loading. At this time, the main damage was the original defect of the sample, the crack of the resin matrix with very thin surface of the material and the degumming of the weak bonding interface. At the stage of microcrack propagation, the energy parameters of the AE signal fluctuated only slightly. This indicated that the damage development of sample was gradual. Because the change of AE energy parameters was basically stable, the damage types can be judged to be basically the same. The damage was mainly caused by matrix crack propagation and debonding at the interface between fiber bundles and matrix. In the stage of instability and failure, the cumulative energy of the acoustic emission signal changed abruptly. At this time, the failure mode was fiber bundle breaking, accompanied by fiber bundle pulling out, which leaded to the final failure of the sample.

### 2.3. HHT Analysis of Acoustic Emission Signals

Samples produced different AE signals at different damage stages. These AE signals at different damage stages tended to overlap and presented random and non-stationary distribution [35]. This brought difficulties to the recognition of AE signals. The Hilbert–Huang transform (HHT) is suitable for the analysis of non-linear and non-stationary acoustic emission signals [36,37]. In this paper, HHT was used to analyze the time-frequency characteristics of acoustic emission signals, and the mode expression of damage evolution process of carbon fiber braided composites was obtained. The HHT method consists of empirical mode decomposition (EMD) and the Hilbert transform [38]. The HHT spectra of AE signals of carbon fiber braided composites were shown in Figure 4.

In Figure 4, the energy of the AE signal was mainly concentrated below 5kHz. With the increase of time, the change of frequency showed some oscillation. The vibration of AE signal wave increases with the increase of frequency. In the initial stage of micro-crack generation (damage initiation), the acoustic signals in the three stages of micro-crack propagation (damage evolution) and instability failure (damage destruction) had obvious burst characteristics. In view of the changes of these acoustic signals, the location of the tensile damage source can be further located.

## 3. Preliminary Positioning by Four-Point Arc Method

According to the HHT time-frequency characteristics of AE signals of carbon fiber braided composites in tensile damage mode, on the basis of determining the damage stage and signal type, a two-step method was used to locate the damage source. In the first step, the sound velocity was compensated by the time-frequency analysis results and the anisotropy of the sample. The initial location was performed by the four-point circular arc method. In the second step, after the initial damage location, the Drosophila algorithm in probabilistic neural network was used to calculate the damage location accurately. The Drosophila optimization algorithm was improved by ergodicity of chaotic algorithm and congestion adjustment mechanism of fish swarm algorithm. In this way, the smoothing parameters of probabilistic neural network were optimized, the number of iterations was reduced, and the accuracy of locating the fixed damage source was further improved.

### 3.1. Acoustic Velocity Correction

Through the analysis of the tensile process and HHT marginal spectrum of carbon fiber braided composites, it can be seen that the acoustic signals of the micro-crack generation stage and the micro-crack propagation stage had obvious continuity characteristics during the tensile process. The acoustic signals in the instability damage stage had obvious sudden characteristics. A number of sensors were fixed on the surface of the object to form a certain geometric relationship, so that the damage source can be initially located by using the time difference of different acoustic waves received by the sensor array. Therefore, the four-point circular arc location algorithm based on the two-dimensional plane location method was used to locate the damage source preliminarily. 

Four acoustic emission sensors were placed on the sample to form a rectangular plane detection area. Because of the anisotropy of carbon fibre braided composites, the acoustic velocity varies greatly, so it is necessary to correct the acoustic velocity to eliminate or reduce the location error caused by the difference of acoustic velocity [39]. Taking the origin as the source of acoustic emission, the lead-breaking experiment was used to calculate the time when the acoustic wave arrived at the sensors. Then, based on the set sensor 1, the time difference of $t_{2}^{\prime }$, $t_{3}^{\prime }$, $t_{4}^{\prime }$, was calculated. After many measurements, the average value of each time difference was taken as a correction parameter. The corrected values of acoustic velocity were shown in Table 1. In the subsequent location calculation, the arrival time of the acoustic wave was corrected by using the modified parameters.

### 3.2. Four-Point Arc Location of Damage Source

Lead breaking experiments were carried out on the samples. The sample was divided into four quadrants by equal ratio. Two points in each quadrant were taken and 10 lead-breaking experiments were carried out at the same point to correct the sound velocity. The actual location of the acoustic emission source was compared with the calculated location of four-point circular arc location by taking the experimental results of eight different locations. The comparison results are shown in Table 2.

As can be seen from Table 2, the accuracy of the four-point arc location algorithm was low, and the error range was about ±10%. Moreover, the closer to the boundary position of the experimental specimen, the calculation error was larger. On the one hand, because of the heterogeneity of the space structure of carbon fiber braided composites, the detection effect was not good. On the other hand, the attenuation of wave velocity and the internal structure of the specimen were affected by the positioning accuracy. It was necessary to improve the location accuracy of acoustic emission source of carbon fiber braided composites by theoretical method.

## 4. Accurate Location of Damage Source Based on Probabilistic Neural Network with Optimized Drosophila Algorithms

Due to the anisotropy of carbon fiber braided composites, the propagation law of acoustic wave in materials was complex, which made it difficult to locate and detect AE damage. In this paper, Drosophila optimization algorithm was selected to optimize the key parameters of the network, taking full account of the propagation medium material, the geometry of the propagation medium, the interweaving of fibers in three-dimensional space and the convergence accuracy and speed of different intelligent optimization algorithms in the calculation of probabilistic neural networks.

### 4.1. Improved Drosophila Optimization Algorithm

The optimization mechanism of the Drosophila optimization algorithm is to simulate the process of Drosophila foraging, which is divided into two stages: olfactory search and visual location. The mechanism of Drosophila optimization algorithm is simple; it has small computational complexity and is easy to implement, and the population can quickly approach the optimal individual, thus ensuring the fast convergence of the algorithm. However, because Drosophila individuals focus on the optimal individuals for random search, it is easy to lead to the premature phenomenon of the algorithm, which reduces the global search ability of the algorithm. Therefore, based on the Drosophila optimization algorithm, combined with ergodicity of chaos algorithm and congestion adjustment mechanism of fish swarm algorithm, an improved Drosophila hybrid optimization algorithm was proposed to locate the damage source of carbon fiber braided composites.

When the chaos optimization algorithm is used to improve the shortcomings of random search, chaos generation mechanism can directly replace random number generator mechanism to improve the efficiency of random search. Therefore, the chaotic search method is used to improve the random search method of Drosophila individuals in order to improve the Drosophila optimization algorithm. Logistic mapping is one of the most commonly used chaos generation mechanisms, which has the characteristics of simple form. This paper chose logistic map as the mechanism of chaotic sequence occurrence, as shown in formula (1):(1)$$z_{n+1}=\mu \cdot z_{n}\cdot (1-z_{n})$$

When the parameter *μ* = 4, the map is full of chaos between 0 and 1. Its chaotic bifurcation diagram and corresponding Lyapunov exponent are shown in Figure 5. As can be seen from Figure 5, with the increase of *μ*, the iteration value of Logistic map goes through a period-doubling bifurcation process of one, two, and four periods, and finally enters into chaotic state. When the logistic map is in chaotic state, its corresponding Lyapunov exponent is larger than 0. The advantages of randomness and ergodicity of chaotic sequence make its search efficiency much higher than that of random search method. Logistic mapping was used to generate the chaotic sequence, which replaced the random sequence generated by a random number generator, so as to realize chaotic search.

In Drosophila algorithm, all Drosophila have the same behavior criteria and aggregate near the food source. As a result, Drosophila cannot search globally, which reduces the global search performance of the algorithm. In the fish swarm algorithm, similar behavior exists in the foraging process, so the concept of crowding degree is set up. That is to say, the fish cannot gather too much in the same place to prevent the algorithm from convergence too fast. In this paper, the concept of crowding degree in the fish swarm algorithm was introduced into the Drosophila algorithm, so that Drosophila did not have the only criterion of action. The optimal initial position is paid according to random probability. Drosophila individuals search around (*x*_best, *y*_best) with probability, and search randomly with small probability.

Therefore, the improvement measures of Drosophila optimization algorithm were as follows:

Step 1—initialization parameters: The main parameters involved in Drosophila optimization algorithm were initialized. Specifically, it includes the maximum number of iterations (MaxGen), the size of Drosophila population (Size), the initial optimal location of drosophila (*x*_*best*, *y*_*best*), and the initial congestion probability (P).

Step 2—setting up chaos generation mechanism: It was shown in formula (2). After selecting the initial value, the chaotic initial value of Drosophila optimization algorithm was obtained after 2000 iterations, which can eliminate the influence of the initial value of chaotic mechanism on the search process.
(2)$$z_{n+1}=\mu \cdot z_{n}\cdot (1-z_{n})$$

Step 3—setting random search direction for Drosophila: Random initial *x*_axis and *y*_axis were set. There were two ways to set the search direction for the *i*th Drosophila individual in the population. Random number generator was used to generate random number *r* between 0 and 1. If *r* < *p*, set the random search direction of the *i*th Drosophila near the locations of *x_axis* and *y_axis*. It was shown in formula (3): (3)$$\begin{array}{l} z_{n+1}=\mu \cdot z_{n}\cdot (1-z_{n})\\ z_{n}=z_{n+1}\\ z_{n+1}=\mu \cdot z_{n}\cdot (1-z_{n})\\ x(i)=x\_axis+z_{n}\\ y(i)=y\_axis+z_{n} \end{array}$$

If *r* ≥ *p*, the random search direction of the *i*th. 

Drosophila was set near the positions of *x_best* and *y_best*. It was shown in formula (4):(4)$$\begin{array}{l} z_{n+1}=\mu \cdot z_{n}\cdot (1-z_{n})\\ x(i)=x\_best+z_{n}\\ y(i)=y\_best+z_{n} \end{array}$$

Step 4—calculated concentration determination value: The distance between the location and the point of the *i*-th Drosophila individual was calculated.

The distance (Dist*_i_*) between the location and the point of the first Drosophila individual is calculated, as shown in formula (5):(5)$$Dist_{i}=\sqrt{{\left(x(i)\right)}^{2}+{\left(y(i)\right)}^{2}}$$

According to the distance, the concentration determination value *S_i_* was calculated, as shown in formula (6):(6)$$S_{i}=1/Dist_{i}$$

Step 5—calculating the concentration (Smell(*i*)) of the *i*th Drosophila: The concentration evaluation function was the optimization function, which was set as *fit*(*S_i_*), the relationship of Smell(*i*) and *fit*(*S_i_*), as shown in formula (7):(7)$$Smell(i)=fit(S_{i})$$

Step 6—preserving the current optimal individuals: Drosophila with the highest odor concentration will be retained as the current optimal individual, as shown in formula (8):(8)[bestSmell,bestindex]=min(smell(i))

Step 7—preserve the optimal concentration and coordinates: The concentration determination values (bestSmell) and position coordinates of the optimal individual were preserved. Drosophila were directed to fly quickly to that location depending on visual perception, which was shown in formula (9):(9)$$\left\{ \begin{array}{l} Smellbest=bestSmell\\ x\_best=x(bestindex)\\ y\_best=y(bestindex) \end{array}\right.$$

The coordinates (*x_best*, *y_best*) were taken as the optimal initial position for the next optimization, and the probability of congestion degree was reduced, as shown in formula (10):(10)p=λ⋅p
In which, 0 < *λ* < 1, *λ* is the probability attenuation coefficient of congestion degree.

Step 8—iterative optimization: Steps 3 to 6 were repeated, and the current optimal concentration was determined to be updated. If so, Step 7 was performed; if not, Steps 3 to 6 were iterated directly. The specified number of iterations was reached, or the algorithm was converged.

In the improved Drosophila optimization algorithm, through the congestion adjustment mechanism of fish swarm algorithm, the initial phase of the congestion probability can be set as a larger probability value, which ensured that Drosophila individuals can carry out sufficient random search to find the optimal location in the global scope. This increased the dispersion of Drosophila individuals and overcame the phenomenon of early maturity in the population. At the same time, the ergodicity of the chaotic algorithm improved the search speed of the algorithm, reduced the number of repeated searches, and thus improved the search efficiency. As the search process proceeds, the probability of congestion decreased gradually. The probability of Drosophila individuals aggregating to the optimal location increased, and the search times near the optimal location increased, which ensured that the algorithm had good convergence characteristics and stability.

### 4.2. Performance Analysis of Improved Algorithms

The standard Drosophila optimization algorithm and the improved Drosophila optimization algorithm proposed in this paper were used to optimize the typical test function to verify the effectiveness of the improved algorithm. The optimization performance of the improved Drosophila optimization algorithm is shown in Table 3.

From Table 3, it can be seen that the improved Drosophila optimization algorithm had a higher optimization efficiency than the standard Drosophila optimization algorithm. In the optimization calculation, the improved function can search the global optimal solution more accurately and obtain better optimization performance. The average error index performance of the optimization results of each function improved. This showed that the method in this paper had stronger global optimization ability. The improved Drosophila optimization algorithm had good optimization performance. Using the improved Drosophila optimization algorithm to optimize the key parameters of the probabilistic neural network can improve the pattern recognition ability of the probabilistic neural network.

### 4.3. Accurate Location of Damage Sources in Carbon Fiber Braided Composites

For the AE detection of carbon fiber braided composites, the output waveform of the sensor was very complex due to the influence of the characteristics of the acoustic emission source, the propagation path of the signal, the environmental noise and the measurement system, etc. If waveform information such as amplitude, rise time and duration was directly used as input, the convergence speed of the network and the accuracy of the output results would not be very high. Therefore, the initial positioning coordinates calculated from the first four-point arc positioning were input as the precise positioning values of the probabilistic neural network damage source of the second optimization Drosophila algorithm, so as to improve the convergence speed and positioning accuracy.

The selection of smoothing coefficients directly affected the performance of probabilistic neural networks. If the smoothing coefficient was too small, the network mainly isolated the training samples, and the probabilistic neural network was equivalent to the nearest neighbor classifier. If the selection of smoothing coefficient was too large, the discrimination of training samples was not large, and the probabilistic neural network was equivalent to the linear classifier. Therefore, the smoothing coefficient directly affects the information processing performance of the network. Because the improved Drosophila optimization algorithm had good global optimization performance, this paper used the improved Drosophila optimization algorithm to optimize the smoothing coefficient of the probabilistic neural network to improve the performance of the probabilistic neural network.

(1) Construction of Optimal Function

For probabilistic neural networks, the selection of appropriate smoothing coefficients was to improve the network’s ability to classify samples correctly. Therefore, the fitness function (optimization function) for solving the smoothing coefficients of probabilistic neural networks by using the optimization algorithm was set as follows (11):(11)$$fitness=f(\sigma_{1},\sigma_{2},\ldots ,\sigma_{s})=\frac{\mathrm{Predict}\;\mathrm{the}\;\mathrm{correct}\;\mathrm{number}\;\mathrm{of}\;\mathrm{samples}}{\mathrm{Total}\;\mathrm{sample}\;\mathrm{number}}$$

For known samples, training samples and test samples were divided. The training samples were only used to construct or determine the structure of probabilistic neural networks, not to optimize the smooth coefficients of networks. The test samples were not used for network construction, but only for optimizing network parameters and evaluating network performance.

(2) Parameter optimization of probabilistic neural network based on the improved Drosophila optimization algorithm

The specific steps of optimizing probabilistic neural network with the improved Drosophila optimization algorithm proposed in this paper were as follows:

Step 1—constructing probabilistic neural network: According to the specific problems to be solved and the number of known samples, a probabilistic neural network model was constructed. If the number of modes was *s*, the parameters to be optimized for probabilistic neural networks were determined to be *σ*_1_, *σ*_2_, … *σ_s_*. In the definition domain of optimization parameters, optimization parameters were initialized

Step 2—setting parameters of the Drosophila optimization algorithm: According to the complexity of probabilistic neural network, the optimum algorithm (MaxGen) was chosen to maximize the number of iterations and the size of Drosophila population. Optimal location parameters and crowding probability values of Drosophila optimization algorithm were set.

Step 3—setting up the mechanism of chaos generation: logistic mapping was selected as the generation mechanism of chaotic sequence, and 2000 iterations are carried out to eliminate the influence of initial value selection on the optimization results.

Step 4—random search of Drosophila: Drosophila individuals are used for chaotic search.

Step 5—calculated concentration determination value: The concentration determination value was calculated.

Step 6—calculating individual concentration of Drosophila melanogaster: The fitness function value was obtained to calculate the concentration value of Drosophila individual’s position.

Step 7—preserving the current optimal individual: Drosophila with the highest odor concentration was retained as the current optimal individual.

Step 8—preserving optimal concentration and coordinates: The concentration determination value (bestSmell) of the optimal individual and its location coordinates were preserved, and the probability attenuation coefficient of congestion degree was adjusted adaptively.

Step 9—iterative optimization: Steps 4 to 7 were repeated to determine whether the current optimal concentration is updated or not. The specified number of iterations was reached, or the algorithm was converged.

The flow chart of optimizing probabilistic neural network with improved Drosophila optimization algorithm was shown in Figure 6.

On the basis of four-point arc preliminary positioning, the initial position is used as the input of probabilistic neural network. The standard Drosophila optimization algorithm and the improved Drosophila optimization algorithm were used to locate the damage source. The results of damage source localization are shown in Table 4.

As can be seen from Table 4, the error range of positioning accuracy based on probabilistic neural network and standard Drosophila optimization algorithm was about ±5%, after using the improved Drosophila optimization algorithm, the positioning accuracy error was about ±1%. This showed that the “two-step method” had a significant improvement in positioning accuracy compared with the traditional single TDOA positioning method.

Furthermore, the HHT spectra of acoustic emission signals of carbon fiber braided composites (Figure 4) were used to locate the damage at each stage. The localization diagram of the damage stage in the tensile test was shown in Figure 7. As can be seen from Figure 7, the brittle breakpoint of the fibers can be seen at the final break fracture. This was also the initial location of the internal damage of the sample. With the increase of tensile load, damage cracks grew rapidly. Two damage points can be located in the damage evolution stage and the damage failure stage. Combining with the location of each point, the damage area of the sample can be basically locked, and the location results of each stage appear near the fracture surface.

By transforming the problem of acoustic emission damage signal identification of carbon fiber braided composites into a mathematical problem of objective function optimization, the problem of damage source localization of three-dimensional braided composites was solved based on “two-step method”. In order to overcome the shortcomings of the Drosophila algorithm in identifying the key parameters of probabilistic neural network, the algorithm was improved based on the idea of chaos optimization and fish swarm optimization. Through numerical calculation and experimental damage source location of different specimens, the results showed that the improved method had significantly improved the identification accuracy of different damage conditions. The optimized probabilistic neural network can locate the damage source accurately in each damage stage of tensile test. In this paper, an improved damage location method was proposed to reduce the error of the conventional location method. This method can locate the damage of carbon fiber braided composite materials in time and accurately, and improve the efficiency of nondestructive testing of composite materials.

## 5. Conclusions

Through this study, the internal damage signal of carbon fiber braided composites under tensile load can be obtained by acoustic emission detection technology. The time domain and frequency domain characteristics of AE signals were analyzed by HHT, and the corresponding relationship between tensile damage mode and AE signal was established. In the first step, the sound velocity was compensated by combining the time-frequency analysis results with the anisotropy of the sample, and the initial localization was carried out by using the four-point arc method of time difference localization. In the second step, probabilistic neural network combined with the standard Drosophila optimization algorithm was used to locate the target in an error range of about ±5%. The construction of smoothing parameters and functions in probabilistic neural network was optimized to reduce the number of iterations. The positioning accuracy error of the improved Drosophila optimization algorithm was about ±1% and the positioning accuracy was improved.

## Figures and Tables

**Figure 1 molecules-24-03524-f001:**
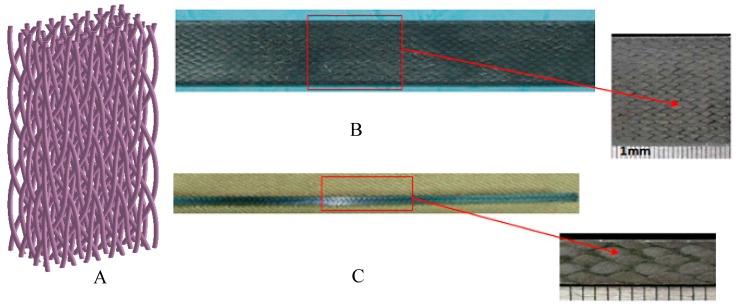
Three-dimensional four-directional carbon fiber braided composites: (**A**) Structural sketch of internal reinforcement; (**B**) Surface morphology of carbon fiber braided composites; (**C**) Side morphology of carbon fiber braided composites.

**Figure 2 molecules-24-03524-f002:**
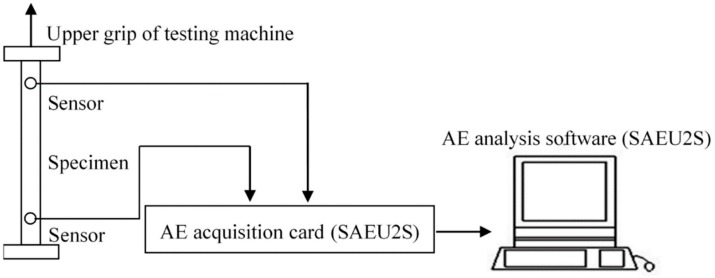
The tensile process of the 3D braided composites.

**Figure 3 molecules-24-03524-f003:**
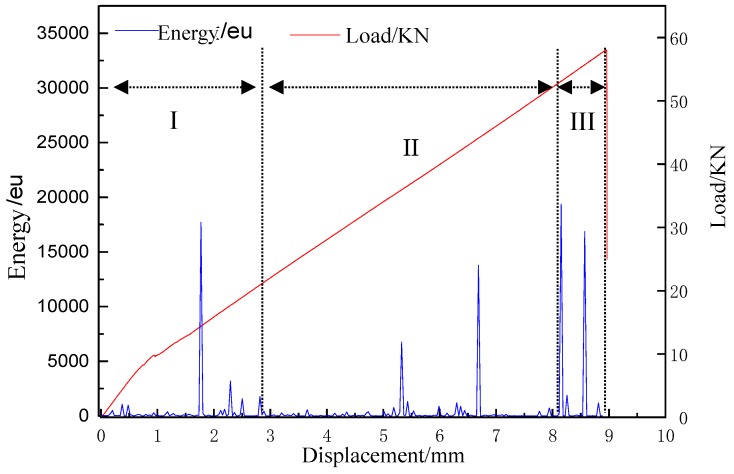
Change in the load-displacement curve corresponding with the acoustic emission (AE) ignal energy parameter of carbon fiber braided composites.

**Figure 4 molecules-24-03524-f004:**
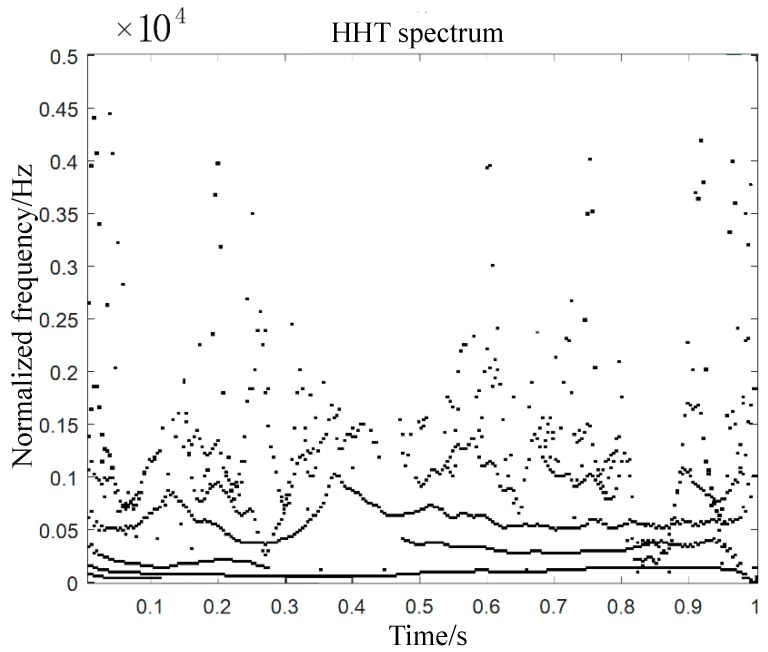
AE signal Hilbert–Huang transform (HHT) spectrum of carbon fiber braided composites.

**Figure 5 molecules-24-03524-f005:**
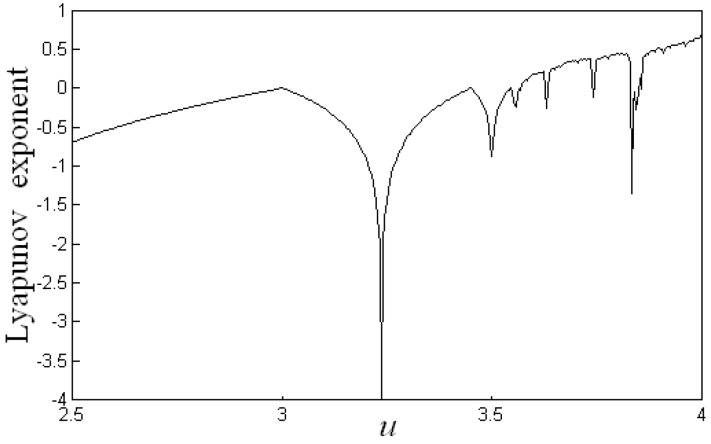
Logistic mapping Lyapunov exponential graph.

**Figure 6 molecules-24-03524-f006:**
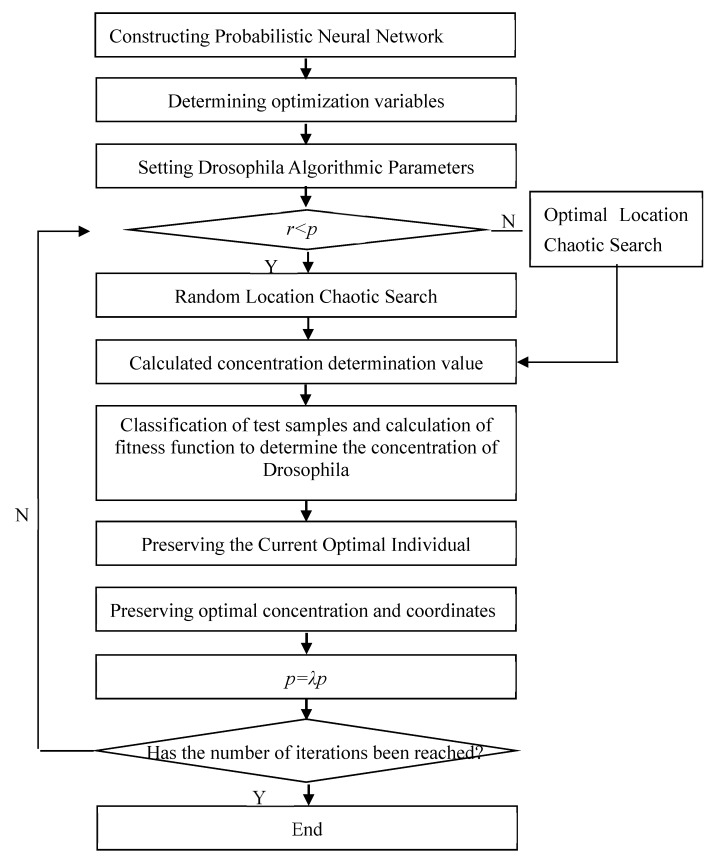
Optimization of probabilistic neural network based on improved fruit fly optimization algorithm.

**Figure 7 molecules-24-03524-f007:**
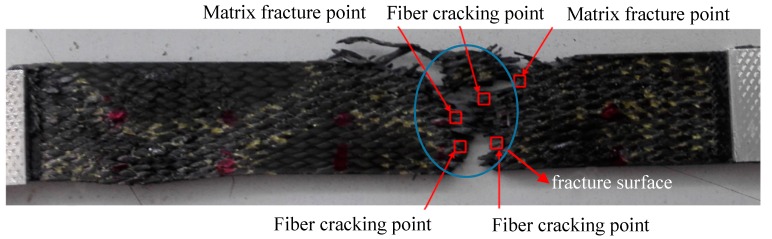
Location of Damage Source in Tensile Test.

**Table 1 molecules-24-03524-t001:** Corrected parameters of acoustic velocity.

Sensor	Velocity (10^3^m/s)	Relative Distance(mm)	Relative Time (s)	Modified Parameters
Sensor 1	5.143	25	4.861	0
Sensor 2	5.069	25	4.932	0.071 × 10^−6^
Sensor 3	4.781	25	5.229	0.368 × 10^−6^
Sensor 4	4.826	25	5.180	0.319 × 10^−6^

**Table 2 molecules-24-03524-t002:** Comparison of the actual location of AE source with the calculated location of four-point circular arc.

Lead Break Location (mm)		Calculated Location (mm)		Error (%)	
X	Y	X	Y	X	Y
3.000	4.000	3.296	4.334	9.857	8.346
5.000	3.000	5.463	3.239	9.267	7.953
−3.000	5.000	−3.270	5.489	8.998	9.785
−6.000	7.000	6.554	7.660	9.231	9.428
−5.000	−6.000	−5.469	6.503	9.389	8.389
−8.000	−4.000	−8.679	−4.382	8.482	9.542
6.000	−9.000	6.590	−9.890	9.833	9.889
10.000	−8.000	11.012	−8.820	10.123	10.251

**Table 3 molecules-24-03524-t003:** Improvement of optimization performance of Drosophila optimization algorithm.

Function	Optimal Point	Global Extremum	Average Absolute Error of Global Extremum	
			Standard Drosophila Optimization Algorithm	Improved Drosophila Optimization Algorithm
*F* _1_	(1.0, 1.0)	0.0	0.0031	0.0002
*F* _2_	(0.0, −1)	3.0	0.0574	0.0261
*F* _3_	(−31.9783, −31.9783)	0.998004	0.1717	0.0862
*F* _4_	(0.0, 0.0)	−1.0	0.0279	0.0086
*F* _5_	(−0.0898, 0.7126) (0.0898, −0.7126)	−1.031628	0.087	0.0139
*F* _6_	(0.0, 0.0)	0.0	0.0548	0.0102

**Table 4 molecules-24-03524-t004:** Comparison of locations of tensile damage sources.

Lead Break Location (mm) (x, y)	Standard Drosophila Computational Position (mm)	Error (%)	Improved Drosophila Computational Position (mm)	Error (%)
	(x, y)		(x, y)	
(1.000, 0.500)	(1.053, 0.525)	(5.345, 4.967)	(1.010, 0.505)	(1.012, 0.934)
(3.000, 1.000)	(2.880, 1.030)	(−3.987, 4.023)	(2.970, 1.009)	(−0.986, 0.894)
(−1.000, 0.800)	(−1.052, 0.762)	(5.167, −4.793)	(−1.011, 0.793)	(1.078, −0.925)
(−6.000, 1.200)	(−5.726 1.172)	(−4.568, −3.368)	(−5.946, 1.188)	(−0.899, −0.967)
(−5.000, −0.700)	(−5.248, −0.733)	(4.962, 4.678)	(−5.052, −0.707)	(1.038, 0.978)
(−7.000, −0.900)	(−6.782, −0.945)	(−3.109, 3.991)	(−6.932, −0.910)	(−0.971, 1.058)
(6.000, −1.000)	(6.254, −1.047)	(4.239, 4.725)	(6.053, −1.009)	(0.891, 0.898)
(4.000, −0.600)	(3.840, −0.569)	(−3.993, −4.234)	(3.957, −0.594)	(−1.079, −0.983)

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
