# Peer review of "Location of Tensile Damage Source of Carbon Fiber Braided Composites Based on Two-Step Method"

_molecules, 2019, doi:10.3390/molecules24193524_

Round 1

Reviewer 1 Report

This is a well written and interesting manuscript with a topic falling within damage location in composite materials, in particular, damage location using acoustic emission testing and a two-step method to locate the damage source. The manuscript reports the methodology to fabricate carbon fiber braided composites, and the two step method to locate the damage. This is an interesting piece of work with some novel results. The results are clear and well discussed; however, the Introduction and methodology sections need improvement as some of the explanations are not very clear. The following comments should be addressed before it can be recommended for publication:

Introduction

In the first paragraph, to put the research in context, the authors should briefly discuss other nondestructive testing techniques for damage location in carbon fiber composites. I recommend to add the following references:

Infrared thermography

https://doi.org/10.1016/j.compstruct.2015.08.119

Ultrasonic C-scan technique:

https://doi.org/10.1016/j.compositesb.2011.02.013

Eddy current testing

https://doi.org/10.1016/j.compositesb.2015.09.041

The authors should then introduce Acoustic emission (AE) technique and mention some advantages of this technique for the particular case of carbon fiber composites.

In the third paragraph of the introduction, the authors mentioned some of the problems of the use of AE for damage location and then they mentioned the content of the paper. Before the content of the paper, the authors should discuss at least 3-4 references regarding the use of optimization algorithms (or other type) and neural network to improve damage location in composites. I recommend to add the following reference (the authors should add 3-4 more):

https://doi.org/10.1177/1475921715607408

In the third paragraph of the introduction, the authors should provide at least some background with some references of the mentioned some of the Chaos algorithm, fish swarm algorithm and the Drosophila algorithm, and some examples of the use of these algorithms in composite materials.

-In the third paragraph of the introduction, the authors should emphasize what is new and unique is this work that has not been done before. i.e. what is the novelty?

Section 2

-In Section 2.2, “Figure 2” should be “Figure 3”.

-In Section 2.2 and Figure 3 caption, “I, II and III sections” should be defined.

Section 4

At the end of Section 4, the authors should add a discussion paragraph emphasizing the advantages and disadvantages of the propose two-step method and discuss other possible applications for composite materials.

Reviewer 2 Report

1、After Drosophila optimization algorithm is improved, we can see that the positioning accuracy error is reduced. How to verify the accuracy of the algorithm after optimization algorithm ?

2、The energy of AE signal showed a rapid decrease at 0.85 seconds, and what caused this phenomenon? Please analyze in detail.
